# Recombinant Technologies to Improve Ruminant Production Systems: The Past, Present and Future

**Andres Alfredo Pech-Cervantes** [1],*, **Muhammad Irfan** [2], **Zaira Magdalena Estrada-Reyes** [1] **and Ibukun Michael Ogunade** [3]

1   Agricultural Research Station, Fort Valley State University, Fort Valley, GA 31030, USA; zaira.estradareyes@fvsu.edu
2   Department of Oral Biology, College of Dentistry, University of Florida, Gainesville, FL 32603, USA; mirfan@dental.ufl.edu
3   Division of Animal and Nutritional Sciences, West Virginia University, Morgantown, WV 26505, USA; ibukun.ogunade@mail.wvu.edu
*   Correspondence: Andres.pechcervantes@fvsu.edu

**Abstract:** The use of recombinant technologies has been proposed as an alternative to improve livestock production systems for more than 25 years. However, its effects on animal health and performance have not been described. Thus, understanding the use of recombinant technology could help to improve public acceptance. The objective of this review is to describe the effects of recombinant technologies and proteins on the performance, health status, and rumen fermentation of meat and milk ruminants. The heterologous expression and purification of proteins mainly include eukaryotic and prokaryotic systems like *Escherichia coli* and *Pichia pastoris*. Recombinant hormones have been commercially available since 1992, their effects remarkably improving both the reproductive and productive performance of animals. More recently the use of recombinant antigens and immune cells have proven to be effective in increasing meat and milk production in ruminant production systems. Likewise, the use of recombinant vaccines could help to reduce drug resistance developed by parasites and improve animal health. Recombinant enzymes and probiotics could help to enhance rumen fermentation and animal efficiency. Likewise, the use of recombinant technologies has been extended to the food industry as a strategy to enhance the organoleptic properties of animal-food sources, reduce food waste and mitigate the environmental impact. Despite these promising results, many of these recombinant technologies are still highly experimental. Thus, the feasibility of these technologies should be carefully addressed before implementation. Alternatively, the use of transgenic animals and the development of genome editing technology has expanded the frontiers in science and research. However, their use and implementation depend on complex policies and regulations that are still under development.

**Keywords:** ruminants; recombinant proteins; vaccine; hormone

## 1. Introduction

The increasing demand for animal-source foods in developing countries will peak in 2050 and, consequently, milk and meat production must double to meet the needs of the population around the world [1]. This effect has pressured the livestock sector to maximize animal production systems, reduce the usage of resources, increase animal performance, and ensure food supply worldwide [2,3]. New technologies for genetic engineering and synthetic biology have been proposed as potential tools for improving living organisms and biological systems [4], including livestock animals. In this context,

ruminant production systems have established a multidisciplinary approach using biotechnology products to improve animal health and performance [5–7].

DNA recombinant technologies have improved the study, characterization, and commercialization of recombinant proteins to improve agro-industrial processes [8,9]. Although biotechnology has fundamentally changed both agricultural and food production recently, the concepts of genetic manipulation and recombinant technology have been discussed for more than 30 years [4,10,11]. Genetic manipulation was initially defined as the formation of new combinations of genetic material by the insertion of nucleic acid molecules using viruses, bacterial plasmids, or other vectors into a host in which those genes are not naturally produced [10]. Modern techniques brought a tremendous improvement in the field of recombinant technology enabling consistent results. Likewise, the progress made in recombinant expression and purification of proteins is under continuous improvement [8,11]. Although many reviews have explained current and previous advances in recombinant proteins, none of these explained the effects of recombinant technologies on the performance and health status of ruminants as a source of food. Thus, the objective of this review is to describe the effects of recombinant technologies and proteins on the performance, health status, and rumen fermentation of meat and milk ruminants.

## 2. Ruminants as a Food Source for Humans and Model of Study of Recombinant Proteins

Domesticated ruminant animals (Figure 1) like dairy cows, beef cattle, sheep, and goats are a group of herbivores that consume plants and non-edible products as a source of carbon and energy [3,12,13]. Energy supply to ruminants depends on enteric fermentation in a multi-chambered stomach and symbiotic relationships with microbes. Thus, understanding the interrelationship between rumen fermentation and rumen microorganisms is key to improving animal performance [10,12]. Moreover, ruminant production systems have important implications for food supply, income, and human health [1,2,12]. Thus, animal production systems like ruminant production systems have increased by 40% in less than 40 years [14]. The rapid intensification of these production systems is required to sustain meat and milk supply by 2030 [9,14]. However, a rapid intensification in ruminant production systems exacerbates the problems associated with animal performance, antibiotic resistance, and food safety [9,15,16]. Consequently, the use of recombinant technologies to improve animal performance, health status, and food supply have been proposed as an immediate alternative [7,11,15]. Despite these promising efforts, there are limitations to adopting recombinant technology in animal production systems. These limitations involve both public and ethical concerns that impose high barriers for marketing and acceptance [4,17]. These limitations depend on the geographical region, regional policies, and economic implications [4,18,19]. For instance, recombinant hormones have been forbidden in the European Union since 1999, however, these are commercially available in the United States and Latin America [19,20]. Although political research suggests that more regulations are required to benefit agriculture from recombinant technologies [17], these regulations will not be described in the present manuscript.

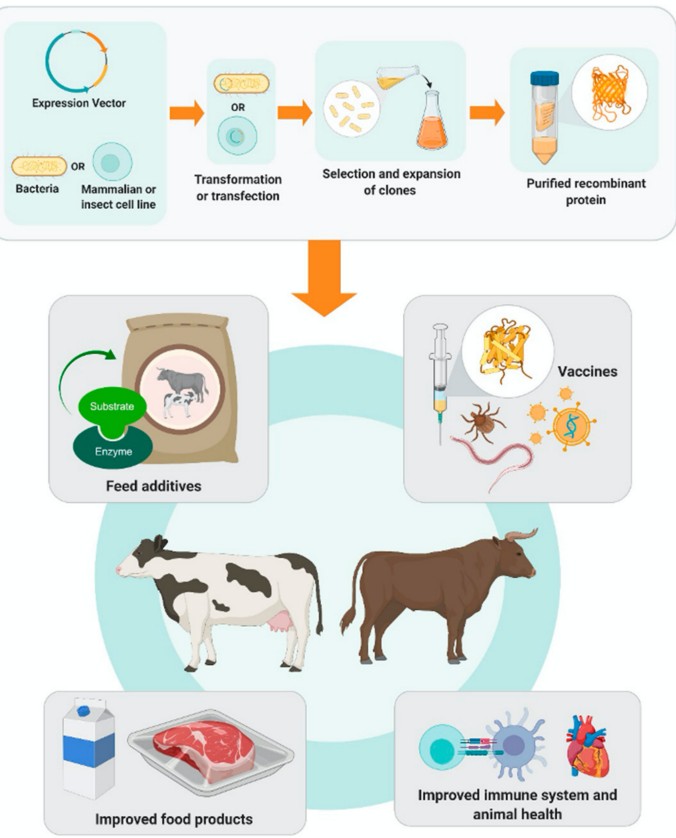

**Figure 1.** Summary of recombinant technologies applied to ruminant production systems.

## 3. Recombinant Hormones for Improving Performance and Fertility

Recombinant hormones have been one of the most widely studied recombinant proteins for improving performance and reproduction in meat and dairy ruminants. The heterologous expression and purification of recombinant bovine somatotropin (rBST) are considered one of the first biotechnology products for application in the livestock industry [5]. Monsanto initially introduced rBST during the early 1990s, three years later in 1993 the Food and Drug Administration (FDA) of the United States approved the use of rBST in dairy and beef cattle [19,21]. The mode of action of rBST includes a cascade of changes in the metabolism of body tissues that increases feed efficiency and milk production [5,20,22]. Previous research demonstrated that rBST improves the immune response of ruminants by increasing the concentrations of IGF-1 in serum but also increases gluconeogenesis in the liver [5,23]. A previous meta-analysis evaluated the effects of rBST on milk production, reproductive performance, and the health status of dairy cows [20,22]. The study showed that the addition of rBST increased milk production by more than 10% in primiparous cows and 15.6% in multiparous cows [22]. Conversely, rBST increased the risk of lameness by 55% [20]. Consequently, the use of rBST was forbidden in the European Union. However, rBST is extensively used in the United States, Mexico, and Brazil for dairy production [19,20,22].

Several recombinant hormones were proposed to improve reproductive performance and immune status in ruminants (Table 1). Previous studies with bovine animals have successfully demonstrated that the use of recombinant gonadotropins improves assisted reproduction in livestock species [24,25]. The recombinant follicle-stimulating hormone (roFSH) is one of the recombinant hormones that promote superovulation in ruminants [26,27]. Exogenous roFSH promotes the maturity of multiple ovarian follicles that improve fertility and the pregnancy rate in ruminants [26,27]. Unlike bacterial expression systems, the heterologous expression of roFSH uses *Pichia pastoris* that enables post-translational modifications resembling those of mammalian cells but not present

in prokaryotic expression systems like *Escherichia coli* [8,27]. Before the development of roFSH, multiple doses of pituitary extracts of FSH were used for the induction of superovulation in cattle with positive results [26,28]. However, recent research demonstrated that a single dose of roFSH was equally effective in inducing superovulation in beef cows and sheep without secondary effects [26]. These results suggested that roFSH could be an alternative to increasing the pregnancy rate in ruminant.

Similar studies with recombinant proteins were conducted by Monsanto during the 1990s including the use of a recombinant bovine placental lactogen (rbPL). Under normal conditions, bovine placental lactogen is a steroid hormone that promotes the growth and development of the mammary gland before lactation [29], and thus rbPL was designed to promote mammary growth and milk production. Similar to rBST, the heterologous expression of rbPL was conducted with *E. coli* BL21. However, compared to rBST, the addition of rbPL did not increase milk yield in dairy cows and did not improve animal performance [29]. These results could be explained by the fact that placental lactogen targets fetal tissues and not the mammary gland [30]. Moreover, similar studies were conducted using retroviral vectors to promote milk synthesis in dairy cows. Compared to bacterial plasmids, viral vectors are more efficient transferring or delivering genes to the cells that impose less technical limitations and allow gene therapy treatments [31,32]. Mehigh et al. [33] used a recombinant leukemia virus vector to deliver a recombinant bovine growth hormone-releasing factor (bGRF) via transfection of bovine cells. The results showed that the recombinant bovine leukemia virus effectively released bGRF in bovine cells 48 h after transfection.

## 4. Recombinant Proteins for Improving Immune Function

The use of recombinant proteins (Table 1) to improve immune function began during the early 1990s. The use of recombinant lysostaphin (rLYS) expressed in *Staphylococcus aureus* and recombinant Interleukin-2 was considered the first therapeutic alternative to reduce mastitis in dairy cows [34,35]. However, the administration of rLYS did not prevent utter infection by *Staphylococcus aureus* compared to the control. Similarly, Interleukin-2 did not prevent udder infections in dairy cows compared to antibiotic treatment [35]. Clinical mastitis is defined as an intramammary infection characterized by a bacterial infection that triggers an inflammatory response and ultimately affects milk production and milk quality [36,37]. In contrast, more recent research reported that transgenic cows secreting lysostaphin in milk prevented the development of mastitis by *Staphylococcus aureus* compared to non-transgenic cows [38]. These results are significant because reducing clinical and subclinical mastitis could save billions of dollars in antibiotic treatments in dairy operations.

The use of recombinant tumor necrosis factor (rbTNF) has been proposed as an alternative to improve immune response and performance of beef and dairy cattle [39,40]. Highly productive animals are subjected to substantial stress from gestation and lactation [40], and consequently tumor necrosis factor -$\alpha$ is endogenously synthesized to prevent inflammation [39]. Thus, the use of rbTNF was proposed to improve the immune response of dairy cattle and increase animal performance after the periparturient period [39]. Although rbTNF improved the immune response of dairy cows with mastitis, the injection of rbTNF promoted the accumulation of triglycerides in the liver that could increase the risk of fatty liver in dairy cows [40]. More recently, the use of a recombinant interleukin-8 (rbIL-8) improved the immune response and performance of dairy cows [15,41]. Endogenous interleukin-8 is secreted by monocytes and other white cells to increase phagocytosis and killing ability [15,41]. Similar to rBST, the heterologous expression of rbIL-8 was conducted using the *E. coli* BL-21 expression system and purified by column affinity. The intravaginal application of rbIL-8 to dairy cows increased blood cell counts, decreased the incidence of puerperal metritis after calving, and improved milk production compared to untreated animals [15,41]. A similarly produced recombinant serum amyloid A3 (M-SAA3) showed antimicrobial activity against pathogenic bacteria isolated from milk infected with mastitis [42]. These studies support the idea that recombinant immune cells could help to improve the health status and performance of ruminants. Moreover, more studies are required to investigate the feasibility of these studies.

**Table 1.** Summary of effects of recombinant hormones and immune cells on the immune status and performance of ruminants.

| Product | Protein | Name | Effect | Expression System | Gene | Mode of Use | Purified | Commercially Available | Results | Author |
|---|---|---|---|---|---|---|---|---|---|---|
| Hormone | Bovine Somatotropin | rBST | Increased feed efficiency and production | *E. coli K-12 and BL21* | Bovine genome | One or more injections | Ys | Yes | Increased milk production, weight gain in both dairy and beef cattle | [5,20–22] |
| Binding protein | Alpha-lactalbumin | alpha-LA | Promote lactose production in dairy cows | Dairy cow | Human alpha-LA | Nuclear transfer of cow embryos | Yes | No | Alpha-LA did not increase lactose concentration in dairy cows | [6] |
| Cytokine | Interleukin-8 | rbIL-8 | Improve immune response in cattle | *E. coli* BL21 | Bovine IL-8 gene | Intravaginal administration | Yes | No | Recombinant rBIL-8 improved the immune response and milk production in dairy cows | [15,41] |
| Hormone | Growth hormone | SbV | Increased muscle deposition | *E. coli* BL21 | Bovine genome | Single injection | Yes | Yes | Increased daily gain and muscle size in beef cattle | [21] |
| Hormone | Follicle stimulatory | roFSH | Superovulatory activity | *Pichia pastoris* | Bovine genome | Single injection | No | Yes | Improve reproductive performance in cattle and sheep | [26] |
| Hormone | Placental Lactogen | rbPL/rbPRL | Improve mammary growth and milk production | *E. coli* BL21 | Bovine genome | Continuous injections | Yes | Yes | Application of rbPL did not increased milk production in dairy cows | [29] |
| Transport protein | Releasing factor | bGRF | Delivery protein system for ruminants | Bovine leukemia virus | Bovine genome | Transfection | Yes | No | The virus infected bovine cells and released bGRF | [33] |

**Table 1.** *Cont.*

| Product | Protein | Name | Effect | Expression System | Gene | Mode of Use | Purified | Commercially Available | Results | Author |
|---|---|---|---|---|---|---|---|---|---|---|
| Protein | Lysostaphin | rLYS | Protein to improve immune response against metritis in cows | *Staphylococcus aureus* | Bovine genome | Injection | Yes | Yes | Application of rLYS did not reduce udder infection in dairy cows | [34] |
| Cytokine | Interleukin-2 | rbIL-2 | Improve immune response in cattle | *E. coli* BL21 | Bovine genome | Intramammary infusion | Yes | No | Application of rbIL-2 was not effective in dairy cows | [35] |
| Cytokine | Tumor necrosis factor | rbTNF | Improve energy metabolism and immune response | *B. brevis* | Bovine genome | Single injection | Yes | Yes | Reduced insulin resistance, improved immune status in heifers | [39,40] |
| Immune cell | amyloid A3 | M-SAA3 | Stimulate innate immunity and prevent udder infections | *E. coli* BL21 | Caprine genome | Incubation in mammary cells | Yes | No | Recombinant M-SAA3 reduced numbers of pathogenic bacteria | [42] |

## 5. Recombinant Vaccines for Improving Immune Response

The rapid intensification of ruminant production systems increased environmental impact and antibiotic resistance [43]. Consequently, there is a growing concern about animal welfare and health that challenges modern ruminant production systems [7,43]. The use of recombinant vaccines provides cost-effective long-term protection against pathogens by stimulating the natural defense system of the host to generate an adequate immune response [7,44,45]. The animal vaccine market is growing to potentially $8.5 billion by 2022 [46]. The field of vaccinology has yielded several effective vaccines that have significantly reduced the impact of some important diseases in both companion animals and livestock [47]. Recombinant vaccines are developed based on rationally designed recombinant highly purified antigens through structure-based design, epitope focusing, or genomic-based screening [44,48]. However, the inherent immunogenicity of recombinant antigens is often low in comparison with the more traditional vaccines, and there is a need for potent and safe vaccine adjuvants to ensure that recombinant vaccines can succeed [45].

Recombinant antigens are commonly combined with adjuvants to enhance immunogenicity (Table 2). The addition of adjuvants to vaccine antigens delivers several advantages, such as dose sparing, increased efficacy in the elderly, and broadening of the cell or/and humoral immune response. Several adjuvants have been evaluated for use in veterinary vaccines, such as mineral salts (aluminum) [48], emulsions (Montanide) [49], and biodegradable polymeric micro-and nanoparticles. Additionally, an alternative range of adjuvants has been described as "immune potentiators" because they exert direct effects on immune cells, thereby leading to their activation. Toll-like receptor (TLR) agonists such as monophosphoryl lipid A [50]; saponins, and bacterial exotoxins are examples of immune potentiators [51]. Some adjuvants such as emulsions in oil, act by sequestering antigens in physically restricted areas, known as depots, to provide long antigenic stimulation. This relatively old-fashioned technology is, nonetheless, a powerful approach that achieves a strong inflammatory response and slow antigen liberation. In contrast to the strongly immune-activating emulsion-type adjuvants, aluminum salt adjuvants are not capable of inducing Th1 or cell-mediated immune activation to any significant degree; however, they are efficient Th2 inducers, giving rise to high antibody titers in the vaccinated individual [47,52].

The DNA vaccines induce antigen production in the host itself. The DNA or RNA vaccine can be defined as a plasmid that contains a viral, bacterial, or parasite gene that can be expressed in mammalian cells or a gene encoding a mammalian protein (non-infectious disease) [47,53]. The gene of interest is inserted into a plasmid along with appropriate genetic elements such as strong eukaryotic promoters for transcriptional control, a polyadenylation signal sequence for stable and effective translation, and a bacterial origin of replication. The plasmid is transfected into host cells and transcribed into mRNA, which is subsequently translated, resulting in the host cellular machinery producing an antigenic protein [47,52]. The host immune system recognizes the expressed proteins as foreign, and this can lead to the development of a cellular and humoral immune response. Immunization of animals with naked DNA encoding protective viral antigens would, in many ways, represent an ideal procedure for viral vaccines because it not only overcomes the safety concerns associated with live vaccines and vector immunity but also promotes the induction of cytotoxic T cells after intracellular expression of the antigens [47].

Subunit vaccines contain short, specific proteins of a pathogen that are noninfectious because they cannot replicate in the host. Protective antigens allow recombinant vaccines to be administered as safe, non-replicating vaccines. There is currently a large amount of scientific interest in the identification of immunogenic and protective antigens for animal pathogens. *E. coli* has been used extensively as a host for heterologous protein expression; however, this approach has some limitations relating to the yield, folding, and posttranslational modifications of the recombinant protein [8]. An alternative host to *E. coli* is the methylotrophic yeast, *Pichia pastoris* [6,54]. This yeast strain has emerged as a powerful and inexpensive expression system for the heterologous production of recombinant proteins that facilitates genetic modifications, allows the secretion of expressed proteins, permits post-translational

modifications, and produces a high yield [55]. The additional benefits of subunit vaccines are that they incorporate proteins in their most native form, thereby facilitating correct protein folding and the reconstitution of conformational epitopes [56]. By incorporating more than one protein into a subunit vaccine, it is possible to invoke immunity to more than one strain or serotype of a bacteria or virus pathogen [47,52]. The potential drawbacks of subunit vaccines are that they offer only a moderate level of immunogenicity and require adjuvants to generate robust immune responses.

The use of recombinant vaccines in cattle production includes the development of "Gavac" against ticks 25 years ago [44]. Cattle ticks (*Rhipicephalus microplus*) are blood-sucking arthropods that affect both humans and animals [44,57]. Conventional methods of control of ticks in ruminants consist mainly of drugs, however, the indiscriminate use of acaricides increased the report of drug resistance around the world [58,59]. Thus, recombinant Bm86 tick protein was expressed in *Pichia pastoris* and used as a recombinant antigen against tick [44]. The results demonstrated that Gavac effectively reduced the proliferation of ticks and reinforced the immune system in cattle, these effects were attributed to the recombinant BM86 protein that disrupted the reproductive capacity of female ticks [44,58].

Similarly, recent research demonstrated that recombinant vaccines effectively protect animals against the most important parasite that affects small ruminants, *Haemonchus contortus*. This gastrointestinal parasite is the major constraint for the ruminant industry in tropical and subtropical areas of the world [60]. The *H. contortus* is a blood-sucking parasite that colonizes the abomasum of small ruminants inducing anemia, reduced weight gain, and weight loss [61]. The hidden antigens include H11 (aminopeptidase H11 glycoprotein) and H-gal-GP (Haemonchus galactose-containing glycoprotein complex) proteins and were initially used to produce recombinant vaccines against *H. contortus* [62,63]. However, the authors suggested that these recombinant vaccines failed to induce a protective immune response mainly due to differences in the glycosylation and conformation between the native and recombinant proteins [63]. Recently, a new recombinant vaccine containing glyceraldehyde-3-phosphate dehydrogenase (Table 2) expressed in *Bacillus subtillis* has shown promising results by decreasing adult worms in the abomasum by 71.5% and increasing weight gain in sheep [64].

The protective effects of recombinant proteins include the development of vaccines against enterotoxaemia, metritis, and bovine diarrhea [65–67]. In this context, Lobato et al. (2010) reported that a recombinant D epsilon toxoid from *Clostridium perfringens* protects cattle and rabbits against enterotoxaemia that causes systemic infections. Likewise, Meira et al. [16] reported that subcutaneous immunization with a recombinant antigen from *E. coli*, *Fusobacterium necrophorum*, and *Trueperella pyogenes* synergistically protected dairy cows against puerperal metritis and improved reproductive performance. Similarly, Jia et al. [66] reported that the use of a recombinant probiotic *Lactobacillus casei* W56 carrying B subunit toxin protected mice against bovine viral diarrhea. In contrast, Otaka et al. [45] reported that a recombinant vaccination did not protect buffalos against botulism, similarly, subsequent immunizations against methanogenic Achaea failed to reduce $CH_4$ emissions in goats [68]. The low immunogenicity frequently observed in recombinant antigens occurs due to a lack of exogenous immune-activating components [47,52]. Recombinant antigens can be offered in different adjuvants, and the immunomodulatory effects are dependent upon the adjuvant used in conjunction with specific antigens and animal species. Thus, the use of recombinant viruses as an alternative to attenuated viruses have shown promising results against the peste-des-pestits virus (PPRV). The PPRV is RNA type virus that transmits a highly contagious disease in both wild and domesticated small ruminants [69,70]. Symptoms of PPRV include high fever, conjunctivitis, ocular and nasal discharges, and diarrhea, consequently, PPRV exhibits mortality from 90 to 100% [53,69,70]. Berhe et al. [53] reported that a dual recombinant vaccine containing a chimera virus against PPRV and capripoxvirus protected goats against PPRV. Similarly, a more recent report, demonstrated that a recombinant Newcastle disease virus (rNDV) protected goats against PPRV [70]. Although the results are promising, the validation of these technologies requires an evaluation on a large scale.

**Table 2.** Summary of effects of recombinant vaccines on the health status of ruminants.

| Product | Protein | Name | Effect | Expression System | Gene | Mode of Use | Purified | Commercially Available | Results | Author |
|---|---|---|---|---|---|---|---|---|---|---|
| Vaccine | Antigen | yidR | Immunity against *Klebsiella pneumoniae* | *E. coli* BL21 | yidR | Immunization using purified protein | Yes | No | ~90% of effectiveness in mice | [16] |
| Vaccine | Antigen | Vrec | Recombinant vaccine against botulism | *E. coli* BL21 | HCBoNT | Immunization using crude extract | No | Yes | Protection for less than 180 days in buffaloes | [45] |
| Vaccine/Protein | Antigen | rBM86 | Vaccine against bovine ticks (*R. Boophilus*) | *Pichia pastoris* MB9 | *BM86 gene present in ticks* | Single injection | Yes | Yes | Provides immune response in domesticated and wild ruminants | [44,57] |
| Vaccine | Antigen | HcGAPDH | Protein against *H. contortus* parasite | *E. coli* BL21/*B. subtillis* | CotB | Single injection | Yes | No | Protective effects against *H. contortus* in sheep | [64] |
| Vaccine | Antigen/Toxin | D-epsion toxin | Vaccine to reduce enterotoxaemia by *clostridium perfringers* | *E. coli* BL21 | ext gene | Immunization using insoluble fraction | No | Yes | It was effective in rabbits and cattle | [65] |
| Vaccine | Antigen/Probiotic | pPG-E20-ctxB | Recombinant vaccine against bovine diarrhea virus | *Lactobacillus* W56 | *V. cholerae* OG80 genome | Direct-fed microbe | Yes | No | Provides immune response in mice | [66] |
| Vaccine | Antigen | LKT/PLO | Vaccine against puerperal metritis in dairy cows | *E. coli* 4612-2 | PLO gene, FimH gene | Subcutaneous injection and intravaginal | Yes | Yes | Increased the immune response increasing lgG titers | [67] |
| Vaccine | Antigen | EhaF | Reduce methanogenesis | *E. coli* DE3 | KP453861 | Intradermal vaccination | Yes | No | The vaccine did not reduce $CH_4$ emissions in goats | [68] |

**Table 2.** *Cont.*

| Product | Protein | Name | Effect | Expression System | Gene | Mode of Use | Purified | Commercially Available | Results | Author |
|---|---|---|---|---|---|---|---|---|---|---|
| Vaccine/Virus | Recombinant virus | Recombinant capripoxvirus reCapPPR/F | Chimera virus to protect against PPRV and capripoxvirus infections | Lamb testis cells | TK gene from ca | PPRV F gene and Thymidine kinase from capripoxvirus | Yes | No | The chimera virus protected goats against PPRV and caproxvirus | [53] |
| Vaccine/Virus | Recombinant virus | Recombinant new castle virus rNDV_H | Recombinant vaccine against peste de petits ruminant virus (PPRV) | Chicken embryo fibroblasts | Glycoprotein h from PPRV | Subcutaneous injections | Yes | No | Administration of rNDV provided protection against PPRV | [70] |

## 6. Recombinant Enzymes for Improving Ruminal Fermentation

Forages represent more than 50% of the dietary ration for ruminant animals [71]. Likewise, dietary fiber plays an important role in the rumen to maintain enteric fermentation, performance, and animal health [3,12]. Thus, increasing fiber digestibility represents a substantial contribution to animal production systems. Supplementation of diets with exogenous enzymes to enhance animal performance has been a practice extensively used for decades to increase feed conversion rates [71,72]. However, the efficiency of exogenous fibrolytic enzymes on the forage digestibility and animal performance of ruminants has been inconsistent [72]. Although cellulose and hemicellulose from forages represent the major source of carbohydrates in ruminants, corn starch is the main component of concentrates in ruminant diets [3,54]. Therefore, extensive research has been conducted to evaluate the effects of recombinant enzymes to improve the fermentation of both fiber and starch in the rumen.

There is an extensive array of non-recombinant microorganisms, including bacteria (*E. coli, Bacillus subtilis, and Bacillus licheniformis*), yeast (*P. pastoris*), and fungus (*Trichoderma reesei* and *Aspergillus niger*), used for the production of enzymes with interest in the feed industry [71,73,74]. However, the use of genetic and protein engineering approaches to produce highly active enzymes with increased resistance to temperature and proteolysis (in many cases derived from extremophile microorganisms), could ultimately result in greater stability of the gastrointestinal tract [75–78]. The recombinant strategies that are being explored aim at designing exogenous enzymes that meet the industry requirements, which include high production yields, low production costs, ease to scale-up, high catalytic efficiency, and improved stability under different temperature and pH conditions. However, most of these studies are still underway and further research efforts are required to develop tools for the application of these strategies.

### Xylanases, Beta-Glucanases, and Amylases

Cellulases and xylanases have been used as a biological pretreatment for forages [3]. Most of the xylanases used in the feed industry for enzymatic treatment of animal feed are derived from those naturally produced in fungi and bacteria [73]. Danisco xylanases (Dupont corporation) and Econase XT (ABEnzymes) produced from fungi *T. reesei*, Prozyme 9300 (Dupont corporation) produced from *Trichoderma longibrachiatum*, Ronozyme WX (DSMNovozyme) in *Aspergillus oryzae*, and Hostazym X (Huvepharma) derived from *Trichoderma citrinoviride* are examples of commercially available enzymes. A recombinant xylanase enzyme is commercially available (Belfeed B 1100 MP, Beldem, *B. subtilis*) for animal feeds; however, most of these commercial products are not pure enzymes, but a mixture of different enzymatic activities [73]. Xylanases and cellulases combined with other enzymes like β-glucanases, amylases, proteases, pectinases synergistically increased fiber degradation in the rumen [73,74]. For instance, β-glucanases are enzymes capable of hydrolyzing cellulose to facilitate the bioconversion of cellulose to animal products. In contrast, α-amylases have been implemented in dairy cattle to improve starch digestibility and milk production [74,79]. Recombinant α-amylases from *B. licheniformis* (Roxazyme® RumistarTM, DSMNovozyme) have been shown to increase feed efficiency and milk production in dairy cows (Table 3) [79].

Given the importance of exogenous enzymes to improve the nutritional value of forages and starch, and the increasing demand for more stable, highly active, and non-expensive carbohydrases, different microbial hosts have been explored for their production [3,74,80]. Although commercial carbohydrases are primarily from fungi, research in this field focuses on the development of bacterial and yeast-based production systems [81]. Yeasts appear as the most promising heterologous expression host for their production as an alternative to fungi. Moreover, some yeast has been accredited a safe status by the FDA, which brings additional value to this expression system. Altogether, these advantages make yeast, and mainly *P. pastoris*, the most widely used microorganism for xylanase production [82]. The *P. pastoris* has been used for the production of xylanases from *T. reesei, Aspergillus sulphureus, A. niger*, and *Streptomyces* sp. *S38*, among others [73]. The Saccharomyces cerevisiae has also been used to produce fungal xylanases. Different enzymes in different yeast-based cell factories have been evaluated

under diverse production conditions aiming to optimize enzyme production yields [83,84]. For instance, the production of catalytically active eukaryotic xylanases in *E. coli* [85]. The *E. coli* expression system has also been used to study different bacterial xylanases [86]. Alternatively, Gram-positive bacteria, like *Lactobacillus* spp. and *B. subtilis* have been used as cell factories for xylanase production purposes. Interestingly, these Gram-positive bacteria have a dual effect, since they are explored as probiotics to enhance gut health, but at the same time, they can secrete recombinant enzymes of interest such as xylanases [87].

Similar to bacterial expression systems, filamentous fungal expression systems (mainly *Aspergillus* spp. and *Trichoderma* spp.) have been also extensively studied for xylanase expression [88]. Other fungi such as *Thermoascus aurantiacus* have also been studied as potential cell factories for xylanase production. Although fungi produce high levels of xylanase, the reduced yield in fermenter conditions and poor secretion efficiency are important to limit factors for their industrial application.

Compared to hormones and vaccines, recombinant exogenous enzymes (Table 3) have been extensively evaluated as an alternative to improve fiber digestibility and animal performance in ruminants [89–92]. Recent studies evaluated the use of recombinant proteins as coadjutants to improve the hydrolysis of fiber and starch. Zein proteins are a group of storage proteins present in the corn endosperm that limits accessibility to starch granules [54], and zein-degrading proteases can improve starch fermentation in corn. This novel recombinant protease has been expressed in *P. pastoris* and can synergistically increase the hydrolysis of corn starch in combination with amylases [54]. Similarly, Pech-Cervantes et al. [77,90] reported that a recombinant expansin-like protein from B. subtilis and fibrolytic enzymes synergistically increased the fermentation of forages in vitro. Expansin-like proteins are a group of non-hydrolytic proteins with disruptive activity towards cellulose. Likewise, Li et al. [76] reported that a recombinant disruptive swollenin from *T. reesei* increased in vitro fermentation of straws. Moreover, Zhang et al. [78] produced a recombinant xylanase-swollenin chimeric enzyme using *P. pastoris*, the enzyme effectively increased hydrolysis of purified cellulose compared to regular xylanase. These results demonstrated that recombinant proteins could enhance rumen fermentation by increasing the hydrolysis of lignocellulose of forages.

Lignin is a polyphenol commonly found in forages that limits cellulose and hemicellulose fermentation in the rumen; therefore, lignification and cross-linkage with polysaccharides reduce the accessibility to forage fiber by rumen microbes [12,13,71]. Alternative strategies to increase the hydrolysis of forages include the use of recombinant laccase enzymes to deconstruct lignin. Liu et al. [93] reported the production of a recombinant laccase from *Lentinula edodes* expressed in *P. pastoris*, and the addition of laccase and cellulase synergistically increased hydrolysis of cellulose and improved lignin deconstruction in straw. Although recombinant enzymes and proteins demonstrated being effective in vitro, very few studies have evaluated these effects in vivo. Ran et al. [91] reported that recombinant xylanase did not improve the digestibility and performance of beef steers. However, Ribeiro et al. [80] reported that a recombinant fibrolytic enzyme increased feed to gain ratio in meat lambs compared to the control. These results imply that recombinant exogenous fibrolytic enzymes could improve animal performance. However, animal variation should be considered as an important factor before implementing these technologies on a large scale.

## 7. Recombinant Direct-Fed Microbes

The use of direct-fed microbes (DFM) as an alternative to growth stimulants, enzymes, and hormones has been proposed in the animal industry for more than 25 years [94]. Probiotics like DFM are live microbial supplements that are capable of altering rumen fermentation and microbiota but also exert immunomodulatory effects on the host [95–97]. Although the effectiveness of non-recombinant DFM has been demonstrated and described [96,98,99], the effects of recombinant DFM have not been described. Table 4 summarizes the effect of recombinant DFM on rumen fermentation, performance, and immune status of ruminants. The expression of fungal enzymes by the recombinant rumen bacterium *Butyrivibrio fibrisolvens* represents one of the first attempts at using recombinant DFM to improve rumen fermentation [10,100,101]. Utt et al. [100] successfully expressed exogenous hemicellulases (xylB gene) from *Aspergillus niger* in a recombinant strain of *B. fibrosolvens* via directional cloning and bacterial transformation. Similarly, Xue et al. [101] functionally expressed recombinant fungal xylanase from *Neocallimastix patriciarum* on both *B. fibrosolvens* OB156 and *E. coli* BL21 strains. Despite these promising results, Kobayashi and Yamamoto [102] reported that recombinant *B. fibrosolvens* were lost after 48 h of incubation in the rumen by protozoa predation. Likewise, Krause et al. [103] reported that recombinant DFM *B. fibrosolvens* did not effectively compete with fibrolytic bacteria in the rumen of sheep. Similar research demonstrated that recombinant bacteria were sensitive to antibacterial factors present in the rumen fluid hence limiting the growth [101,104]

Recombinant *B. fibrosolvens* carrying a dehalogenase gene was able to prevent fluoroacetate poisoning in ruminants [105,106]. Fluoroacetate is a toxic compound present in plants around the world like Australia, Africa, and South America. Domestic animals are commonly killed by fluoroacetate poisoning [105,106]. The use of recombinant fluoroacetate dehalogenase enzyme expressed in *B. fibrosolvens* demonstrated the capacity to prevent poisoning in sheep without side effects [105]. These results showed the effectiveness of recombinant DFM for the first time. Similarly, the use of recombinant yeast to improve fermentation has shown promising results. Several studies have evaluated the effects of recombinant yeast expressing cellulases and amylases to improve rumen fermentation [83,107,108]. Yamakawa et al. [83] reported that recombinant *S. cerevisiae* expressing recombinant alpha-amylase and glucoamylase (Table 4) increased raw starch fermentation compared to a non-recombinant yeast. Selwal et al. [108] reported that recombinant *S. cerevisiae EBY100* strain expressing an alpha-amylase from *Aspergillus niger* increased the release of glucose and maltose from starch at rumen conditions (pH 7 and 40 °C). Furthermore, Haan et al. [107] reported that a recombinant *S. cerevisiae* expressing a beta-glucosidase from *Trichoderma reeesei* increased hydrolysis and fermentation of purified cellulose. Despite these promising results, in vivo studies are required to demonstrate the effectiveness of recombinant DFM on fiber digestibility and rumen fermentation.

## 8. Recombinant Technologies to Improve Sustainability of Animal Food Sources

Recombinant technologies have proven to be effective in improving animal performance and health in ruminants. However, their use has been extended to the food industry as a strategy to enhance the organoleptic properties of animal-food sources, reduce food waste, and mitigate the environmental impact [109–111]. The biotechnological potential of recombinant probiotics, proteins, and enzymes has been explored using both dairy and meat products [109,112,113]. Examples of these technologies include the use of recombinant transglutaminases, chymosin, and galactosidases to improve the quality of meat and dairy products (Table 5).

Transglutaminases are enzymes commonly used to bind mixtures of restructured (ground) meat and minced meat that improves processing and reduce waste. Thus, recombinant transglutaminases expressed in *P. pastoris* increased the quality of restructured meat products compared to the control [113]. Likewise, Yeh et al. [114] previously demonstrated that recombinant *Lactococcus lactis* expressing an antifreeze protein analog-reduced protein loss, drip, and improved the organoleptic properties of frozen meat. Furthermore, Stephan et al. [115] reported that the heterologous expression of a recombinant colicin reduced *E. coli* counts on fresh steaks. Colicins are non-antibiotic bacterial proteins that prevent bacterial enteric infections like Shiga-toxin producing *E. coli* [116]. Similarly, Chen et al. [112] reported that a recombinant β-galactosidase improved milk lactose hydrolysis and reduced the time it took hydrolysis of lactose in milk to improve lactose-free milk production. These results demonstrated that enzymatic pretreatment of both meat and dairy products could reduce food waste and improve food processing.

Chymosins are a group of proteases produced in the abomasum of ruminants that are used to produce cheese curds. Thus, recombinant bovine chymosins have been used to improve cheese production for more than 10 years [117]. Unlike calf or lamb chymosin, recombinant chymosins can be produced on a large scale with low cost and predictable coagulation behavior that increases the efficiency of cheese production [109,117]. More recently, the use of recombinant probiotics has been proposed to reduce the environmental impact of cheese whey [111]. Cheese whey is the serum portion of milk that remains after the cheese-making process. However, cheese whey is one of the most pollutant by-products of the food industry [118]. Consequently, Boumaiza et al. [111] suggested that a recombinant *Lactococcus lactis* bacterium expressing a monellin protein could help to reduce the environmental impact by reintroducing cheese whey as a substrate for recombinant lactic acid bacteria and the potential use of monellin as an industrial sweetener. These results demonstrated that recombinant technologies could help to improve the sustainability of animal-food sources by optimizing the manufacturing process of food and reducing the harmful effects of food by-products. Ultimately, more studies are required to implement the use of recombinant technologies in food production.

## 9. Transgenic Animals

The use of transgenic animals has also been proposed as an alternative to improve animal production systems. More than 30 years ago Murray et al. [119] proposed the use of transgenic sheep with high levels of growth hormone to improve growth and animal performance. The animals were produced by pronuclear microinjection that consists of injecting genetic material into the nucleus of a viable oocyte [120], resulting in transgenic animals with high concentrations of growth hormone in blood and tissues. Five years later, Powell et al. [121] produced transgenic sheep expressing high levels of wool keratin protein, however, the transgenic rate was only 13% from a total of 516 lambs. Similarly, Brophy et al. [122] produced 11 transgenic calves expressing high levels of β- and κ-casein in milk. Casein is a group of proteins present in milk, the concentration, and the type of casein in the milk influences quality [122]. Transgenic dairy cows produced between 8% and 20% more β-casein in the milk compared to non-transgenic animals. Although the transgenic rate was low (from 4% to 20%), the transgenic animals had twice the concentration of the total casein in the milk compared to the non-transgenic animals. These results imply a substantial increase in the quality and properties of milk. More recently, Wang et al. [123] proposed the use of transgenic cows for large-scale production of lactoferrin. Human lactoferrin is a glycoprotein with applications in pharmaceutical products, including cancer treatments [124]. Results from that study showed that transgenic cows produced between 4.5 to 13.6 g/L of recombinant lactoferrin. Thus, transgenic animals exhibited an enhanced immune system due to the therapeutic effects of lactoferrin in the cell [123,124].

The growing challenges associated with ruminant production systems include the presence of diseases, parasites, and metabolic disorders. The use of transgenic technology could be a feasible alternative to improve animal production [125]. In 1986, the epidemic of bovine spongiform encephalopathy impacted beef production in the United Kingdom, and beef products contaminated with prions caused an outbreak of human Creutzfeldt–Jacob disease [126]. Consequently, Richt et al. [127] proposed the use of transgenic cattle engineered by the chromatic transfer procedure with the prion protein (PrPc) disrupted. Compared to pronuclear microinjection, chromatin transfer is a cloning technique that allows the reprogramming or elimination of deleterious genes. The results showed that brain tissue from transgenic animals was resistant to prion propagation. Another study demonstrated that genetically enhanced cattle and goats with the ability to secrete lysostaphin in the milk exhibited protection against intramammary infections by *Staphylococcus. aureus*. These results showed that the production of mastitis-resistant cattle could be used as a potential solution for intramammary infections by *S. aureus* [38,128]. Despite public concerns, transgenic animals could save billions of dollars in antibiotic treatments, animal culling, and economic losses in large-scale animal operations.

## 10. Gene Editing: An Emergent Technology to Improve Animal Production Systems

The summary of effects of recombinant enzymes, recombinant microbes, recombinant protein on ruminants are shown in Tables 3–5. The development of genome editing technology has expanded the frontiers in science and research. Gene editing is a powerful technology that allows an accurate modification of the genome of an organism [129,130]. Technologies for gene editing are divided into three main tools; zinc-finger nucleases (ZFNs), transcription activator-like effector nucleases (TALENs), and cluster regularly interspaced short palindromic repeats/associated nuclease cas9 (CRISPR/cas9) [129–131]. These revolutionary technologies will transform livestock systems by selectively improving animal breeds and controlling gene diversity [130]. Although some of the technologies have been explored experimentally to improve livestock production, none of these technologies have gained approval for commercialization [129,131]. Previous reviews have described the outcomes and the upcoming benefits of using gene editing in livestock production. However, similar to recombinant technologies, consumer groups and the industry sector remain uneasy over gene-editing technology [129,130,132]. New policies and laws are required to regulate the commercialization and distribution of gene editing-based animals. However, unlike recombinant technologies, synthetic biology has forced the creation of a new regulatory system because edited animals or products do not contain traces of recombinant material [17,19,133]. Moreover, controlling access to that technology represents an important challenge in the future.

**Table 3.** Summary of effects of recombinant enzymes on in vitro and in vivo fermentation and performance of ruminants.

| Type | Protein | Name | Effect | Expression System | Gene | Mode of Use | Purified | Commercially Available | Results | Author |
|------|---------|------|--------|-------------------|------|-------------|----------|------------------------|---------|--------|
| Enzyme | Protease | ZDP | Zein-degrading protease | *Pichia pastoris* X-33 | Zeocin gene | Incubation with alpha amylases | Yes | No | Synergistic hydrolysis of starch between ZDP and amylases | [54] |
| Enzyme | Cellulase | pILCT-C | Fungal cellulase in *L. lactis* for silage inoculants | *E. coli* V850 | Neocallimastix fungi genome | Inoculation in silage | No | No | Increased NDF hydrolysis in alfalfa samples | [75] |
| Enzyme-like | Swollenin | pPICZalphaA | Disruptive activity towards cellulose | *Pichia pastoris* X-33 and *E. Coli* DH5alpha | swoF | Purified Swollenin + fibrolytic enzyme were applied to a diet for in vitro digestibility | Yes | No | Increased in vitro fermentation and acetate concentration | [76] |
| Enzyme-like | Expansin-like protein | BsEXLX1 | Increase cellulose and hemicellulose fermentation | *E. coli* BL21 | yoaJ | Direct application to the substrate | Yes | No | Synergistic degradation of fiber with fibrolytic enzymes. Increased rumen fermentation in vitro | [77,90] |
| Enzyme | Xylanase | rLexyn11a | Hydrolytic activity towards hemicellulose | *Pichia pastoris* X-33 | XynR | Direct incubation in the diet for beef cattle | Yes | No | Increased in vitro fermentation and increase acetate and butyrate concentrations | [78] |
| Enzyme | Amylase | Rumistart | Increased hydrolysis of starch in the rumen | *Bacillus licheniformis* | NA | Direct application to the feed | No | Yes | Recombinant amylase increased feed efficiency and milk production in dairy cows | [79] |

**Table 3.** *Cont.*

| Type | Protein | Name | Effect | Expression System | Gene | Mode of Use | Purified | Commercially Available | Results | Author |
|------|---------|------|--------|-------------------|------|-------------|----------|------------------------|---------|--------|
| Enzyme | Celullase-Xylanase | GH10/XYL10A | Increased cellulose and hemicellulose degradation | *E. coli* BL21 | *Aspergillus Niger* genome | Direct application to the substrate | Yes | No | Recombinant enzymes increased degradation of straw in vitro and daily gain in sheep | [80,89,134] |
| Enzyme | Laccase | LeLac | Degradation of lignin from lignocellulose | *E. coli* Bl21 and *P. Pastoris* | *L. edodes* AB035409.1 gene | Direct application to the substrate | Yes | No | Increased lignin degradation in straw | [93] |
| Enzyme | xylanase | XOS | Hydrolysis of hemicellulose | *Pichia pastoris* GS115 | Xyn10CF | Direct incubation in agricultural waste | No | No | Increased hydrolysis of hemicellulose | [135] |
| Enzyme | Cellulase | CMC-1 and EP | Improve fiber fermentation | *E. coli* BL21 | CMC-1, EP-15 | Hydrolysis of cellulose at rumen conditions | Yes | No | High activity towards cellulose at rumen conditions | [136] |
| Enzyme | Amylase | amyB | Increasing hydrolysis of starch | *Bacillus choshinensis* | Amybeta | Crude extract obtained by solid-state fermentation | No | No | Increased glucose release from starch | [137] |
| Enzyme | Xylanase | xynS20 | Increase hemicellulose digestibility in the rumen | *E. coli* BL21 | *N. patriciarum* genome | Direct application to the substrate | Yes | No | Increased hydrolysis of lignocellulose | [138] |

**Table 4.** Summary of effects of recombinant direct-fed microbes (DFM) on the in vitro and in vivo rumen fermentation and health of ruminants.

| Product | Protein | Name | Effect | Expression System | Gene | Mode of Use | Purified | Commercially Available | Results | Author |
|---|---|---|---|---|---|---|---|---|---|---|
| Yeast | Amylase and glycoamlyse | alpha-amylase | Increasing starch fermentation | *S. cerevisiae* MT8-1 (lithium acetate method) | SBA, SBAI, SBAII, SABIII | Incubation in corn | no | no | Increased fermentation of starch compared to the control | [83] |
| Yeast | Amylase | Y294-Amy | Increase starch hydrolysis and fermentation | *S. cerevisiae* Y294 and *E. coli* (lithium acetate method) | apuA, temA, ateG, temG | Direct application to the substrate | Yes | No | Increased starch fermentation | [84] |
| Bacteria | fungal xylanase | xynA | Improving fiber fermentation | *E. coli* BL21 | pNPXD2 | Direct-fed microbe | Yes | No | Low competitiveness with rumen microbes | [101] |
| Bacteria | Xylanase | xynA | Increase fermentation of hemicellulose | *Butyrivibrio fibriosolvens* | xynA, pUMSXr | Direct-fed microbe | Yes | No | Low competitiveness with rumen microbes | [103] |
| Bacteria | Cellulase | rLB pM25 | Increase fiber degradation and fermentation in the rumen | *Lactobacillus Plantarum* | *Clostridium thermocellum* | Direct-fed microbe | Yes | No | The bacteria were rapidly lost by protozoal predation | [104] |
| Yeast | Cellulase | BGL1 | Recombinant *S. cerevisiae* with B-glucosidase and Cellulase | *E. coli* XL1 and *S. cerevisiae* Y294 | X99228, AB003694 | Hydrolysis of purified cellulose | Yes | Yes | Increased hydrolysis of cellulose | [107] |
| Bacteria | Dehalogenase | pBHf | Prevent fluoroacetate poisoning in ruminants | *Butyrivibrio fibriosolvens* | *M. species* | Direct-fed microbe | Yes | Yes | Recombinant bacterium prevented poisoning in sheep | [105,106] |
| Yeast | Amylase | pYDI | Increase starch hydrolysis and fermentation | *E. coli* Bl21 and *S. cerevisiae* | *Aspergillus niger* NRRL334 | Direct incubation in the rumen | Yes | No | Increased starch hydrolysis at rumen conditions | [108] |

**Table 4.** *Cont.*

| Product | Protein | Name | Effect | Expression System | Gene | Mode of Use | Purified | Commercially Available | Results | Author |
|---|---|---|---|---|---|---|---|---|---|---|
| Bacteria | Xylanase | rBTX | Increase hemicellulose digestibility in the rumen | *Bacteroides thetaiotaomicron* | *Prevotella ruminicola* genome | Direct-fed microbe | Yes | No | Did not improve fermentation | [133] |

**Table 5.** Summary of effects of recombinant proteins on processing and sustainability of animal-food sources.

| Type | Protein | Name | Effect | Expression System | Gene | Mode of Use | Purified | Commercially Available | Results | Author |
|---|---|---|---|---|---|---|---|---|---|---|
| Enzyme | Transglutaminase | MTG | Improve meat product quality | *Pichia pastoris* GS115 | TGase gene from *S. fradiae* | Direct application in restructured meat | Yes | No | Direct application of MTG increased meat quality | [113] |
| Protein/bacteria | Antifreeze protein | rAFP expressed by *L. lactis* | Improve cryogenic preservation of meat | *Lactobacillus Acidophilus, Lactoccocus lactis* | SlpA from *L. Acidophilus,* | Direct application of lyophilized crude extract on meat and dough | No | No | Increased juiciness and reduced protein loss in frozen meat. Improved fermentation of dough | [114] |
| Protein | Colicin | ColM | Antibacterial activity for meat and food | *Nicotiana Benthamiana* | Colicin gene from *E. coli* | Direct application on meat | Yes | No | Application of ColM reduced *E. coli* counts on fresh steak meat | [115,116] |
| Enzyme | Chymosin | RLC | Improve milk clotting for cheese production | *E. coli* | Lamb gene | Direct incubation on milk for cheese production | Yes | Yes | Improved cheese production compared to the control | [109,117] |
| Probiotic | Monellin | MNEI | Sweetener from cheese whey | *Lactococcus lactis/E. coli* | MNEI gene | Direct incubation of *L. lactis* on cheese whey | Yes | No | This strategy valorized dairy effluents like cheese whey to produce Sweetener and probiotics | [111] |

**Table 5.** *Cont.*

| Type | Protein | Name | Effect | Expression System | Gene | Mode of Use | Purified | Commercially Available | Results | Author |
|---|---|---|---|---|---|---|---|---|---|---|
| Enzyme | Galactosidase | bgaB | Improve Lactose hydrolysis | *Bacillus subtilis* | *Bacillus Stearothermophilus* gene | Direct incubation on whole-milk for lactose-free milk production | Yes | Yes | Improved hydrolysis of lactose in milk | [112,139] |
| Probiotic | Enzyme | RD-534 | Improve exopolysaccharides in yogurt | *Streptococcus thermophilus* RD534 | *S. thermophilus* DGCC7710 | Direct incubation on milk for yogurt production | Yes | No | Addition of RD-534-S1 increased production of exopolysaccharides | [140] |
| Enzyme | Transglutaminase | TGZo | Food enhancer to produce yogurt | *Pichia pastoris* GS115 | TGZo gene from Corn | Incubation for yogurt production | Yes | No | TGzo increased consistency, cohesiveness and viscosity in yogurt | [141] |

## 11. Conclusions

Over the last 25 years, recombinant technologies greatly improved ruminant production systems by enhancing feed efficiency, reproductive performance, and meat and milk production. However, most of these technologies are still highly experimental implying that in vivo evaluations are required before implementation. Nonetheless, recombinant technologies have proven to be an effective alternative to improving animal performance without compromising animal health. The use of recombinant technologies could reduce the carbon footprint and environmental impact of livestock and reduce food waste; however, the feasibility of these technologies should be evaluated. Across the literature, experimental evidence suggests that recombinant technologies could improve the sustainability of ruminant production systems and ensure food supply. Furthermore, transgenic animals could be an alternative to reduce the use of antibiotics and improve animal performance, however public concerns should be addressed and discussed. Finally, taking advantage of gene editing technologies could help to ensure global food security by proving lower-cost products.

**Author Contributions:** Conceptualization, A.A.P.-C. and M.I.; methodology A.A.P.-C. and M.I.; software, Z.M.E.-R.; validation A.A.P.-C., M.I., I.M.O. and Z.M.E.-R.; formal analysis, A.A.P.-C.; investigation, A.A.P.-C.; resources, A.A.P.-C.; data curation, A.A.P.-C., M.I., I.M.O. and Z.M.E.-R.; writing—original draft preparation, A.A.P.-C. and M.I.; writing—review and editing, A.A.P.-C., M.I., I.M.O. and Z.M.E.-R.; visualization, A.A.P.-C., M.I., I.M.O. and Z.M.E.-R.; supervision, A.A.P.-C., M.I., I.M.O. and Z.M.E.-R.; project administration, A.A.P.-C.; funding acquisition, A.A.P.-C. All authors have read and agreed to the published version of the manuscript.

**Funding:** This research was funded by the department of agriculture of the United States (USDA) grant number 1022336.

**Conflicts of Interest:** The authors declare no conflict of interest.

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
