# Peer review of "Recombinant Technologies to Improve Ruminant Production Systems: The Past, Present and Future"

_processes, doi:10.3390/pr8121633_

Round 1

Reviewer 1 Report

This study is a very well-drafted review article covering the technological and field aspects of recombinant products for enhancing ruminant products. The manuscript has a very sooth flow with few minor grammatical errors and typos. The concepts of the technology in question have all been covered including a concise intro that presents the full scope. Also, immune functions and responses, ruminal fermentation enzymes, ruminant probiotics and finally genomics are comprehensively covered; which give this manuscript significant scientific merit.

However, one key topic is almost completely missing in this manuscript and that is effect of recombinant technologies on ruminant products sustainability. Given the fact that sustainability is indeed the toughest challenge that ruminant products face, the manuscript must embody a section covering this crucial topic.

Author Response

Dear Reviewer,

Thank you for your valuable comments and suggestions. The manuscript was revised and updated accordingly. Following your comment, we added another section that addresses the effects of recombinant technologies on the sustainability of animal food sources. Please see from L434 to 468, a supporting table was included as well.

Sincerely,

The authors

Reviewer 2 Report

Dear authors,

I really appreciate your comprehensive review paper with actual topics. It is written in very well readable style.

The only comments are:

- Titles of all tables are on the separate page (line 124, line 188, line 345, line 445, this probably can be changed in the final form of manuscript -  to have title and table on the same page.

- The numbers of the references are doubled (page 518 and further on, errors in numbering are also in lines 862 - 863,...). Some references are written in CAPITALS. References should be unified to requested style of the journal.

I recommend to accept this manuscript after correction of these minor mistakes at the present form.

Best regards,

Author Response

Dear Reviewer,

AU: Thank you very much for your valuable comments about our manuscript.

AU: Thank you for the comment. The tables were updated and now these are editable and the titles were placed on the same page.

AU: Thank you for the comment. The references were corrected accordingly and the capitals were corrected.

Sincerely,

The authors

Round 2

Reviewer 1 Report

Nicely revised! 

I am satisfied now.